# Prediction of Elastic Properties Using Micromechanics of Polypropylene Composites Mixed with Ultrahigh-Molecular-Weight Polyethylene Fibers

**DOI:** 10.3390/molecules27185752

**Published:** 2022-09-06

**Authors:** Jong-Hwan Yun, Yu-Jae Jeon, Min-Soo Kang

**Affiliations:** 1Regional Innovation Platform Project of Kongju National University, Cheonan-si 31080, Korea; 2Department of Medical Rehabilitation Science, Yeo-ju Institute of Technology, Yeoju-si 12652, Korea; 3Division of Smart Automotive Engineering, Sun Moon University, Asan-si 31460, Korea

**Keywords:** UHMWPE, polypropylene, composite, homogenization, micromechanics

## Abstract

In this study, we calculated the elastic properties of polypropylene composites mixed with ultrahigh-molecular-weight polyethylene (UHMWPE) fibers. We applied micromechanics models that use numerical analysis, conducted finite element analysis using the homogenization method, and comparatively analyzed the characteristics of polypropylene (PP) containing UHMWPE fibers as reinforcement. The results demonstrate that elastic properties improved as the volume fraction of UHMWPE fiber increased. It was confirmed that the fibers had anisotropic elastic properties due to the shape of the fibers. In addition, it is necessary to compare these findings with future experimental results to obtain data for developing UHMWPE–PP composites.

## 1. Introduction

Many modern technologies require materials with unusual combinations of properties that cannot be obtained by conventional metal alloys, ceramics, and polymeric materials. Material property combinations and ranges have been and continue to be extended by the development of composite materials. A composite is a multiphase material that exhibits a significant proportion of the properties of both constituent phases, such that a better combination of properties is realized. According to this principle of combined action, better property combinations are fashioned by a judicious combination of two or more distinct materials. Combinations of these materials largely consist of three types: particle-reinforced, fiber-reinforced, and structural. Of these, continuous synthetic fiber-reinforced polymer (FRP) has excellent properties, including a particularly high strength-to-weight ratio, and thus has been applied in various fields, such as F1 automobiles, sports accessories, aviation and aerospace, and shipbuilding [1,2,3,4]. Fibers are classified as natural or synthetic, depending on their origin. However, each type of fiber has advantages and disadvantages. Synthetic fiber-reinforced thermoplastic composites have better mechanical strength than natural fibers but are not environmentally friendly [5,6,7]. Nevertheless, researchers have performed numerous studies on synthetic fibers due to their excellent mechanical properties. A unidirectional composite with a hexagonal array of fibers can be transversely isotropic because the properties are the same along any plane, which is normal to the fiber direction [8,9]. The stiffness and strength of a unidirectional composite exhibit anisotropic properties as they vary with orientations. The stiffness of unidirectional composites in the fiber direction is usually dominated by the fiber properties while the strength in the transverse direction is dominated by the matrix properties. Many fibers are used in fiber-reinforced composites, with ultrahigh-molecular-weight polyethylene (UHMWPE) particularly garnering attention. UHMWPE fibers possess excellent properties, such as high axial strength, high axial modulus, excellent resistance to chemicals, low dielectric constant (2.25) and dielectric loss tangent (2.0 × 10^−4^), and is not affected by water [10,11]. Due to its low density (0.95 g/cm^3^), UHMWPE fibers exhibit excellent specific strength and specific modulus compared to other reinforcing fibers, such as glass and Kevlar fibers. These excellent mechanical properties of UHMWPE fiber have made it an attractive consideration as reinforcement in high-performance polymer matrix composites. Unidirectional composites have all fibers aligned in a single direction. As described above, fibers are used as reinforcement to improve the strength of composites, and the polypropylene (PP) matrix maintains the desired position and direction and prevents damage to the fibers. PP, a type of polyolefin, is a light and weldable crystalline plastic created through the catalytic polymerization of propene. PP material is a well-balanced engineering plastic with excellent chemical resistance and electrical insulation properties, high purity, low water absorption, and a wide range of applications. As PP is typically made into products via extrusion or thermocompression molding, it is mainly used as a matrix material when developing composites using fiber reinforcement. To develop UHMWPE–PP composites, it is important to compare elastic properties via micromechanics to analyze the behavior of the composite when mixing the UHMWPE fibers with PP. Numerical methods used to calculate the properties of composites generally include the analysis of representative volume elements (RVEs) [12,13,14]. Researchers have proposed many micromechanics models to predict various mechanical properties of composites [15,16,17,18]. Some models, such as numerical homogenization and finite element analysis (FEA), have also been proposed. Rule of mixture (ROM) can linearly predict the elastic properties of composites according to the volume fraction, and the Halpin–Tsai model is a series of empirical equations that can express the properties of composites in terms of reinforcement while considering the properties of the matrices, their ratios, and geometry. This numerical approach is effective for predicting and utilizing the physical behavior of composites. Accordingly, to develop a composite added with UHMWPE fibers, this study comparatively analyzed the changes in elastic properties according to the addition of UHMWPE fibers (2–70% volume fraction) through numerical analysis using FEA and micromechanics.

## 2. Results and Discussion

Based on the FEA and numerical analysis results, the changes in elastic modulus are shown in Figure 1. The results of changes in elastic moduli E1, E2, and E3, calculated according to the volume fraction of UHMWPE fibers, are as follows. The transverse elastic modulus of the fibers is linearly combined with the elastic properties according to volume fraction, linearly expressed as E1 in Figure 1. Thus, the properties of the transverse elastic modulus increased according to the volume fraction of UHMWPE in the numerical analysis results and FEA model, attributed to the large difference in elastic modulus values, as the elastic modulus of UHMWPE is 25,000 MPa and that of PP is 1325 MPa. This result indicates that as the volume fraction of UHMWPE fibers (with a high elastic modulus) increases, the elastic modulus increases in proportion to the volume fraction. However, the longitudinal elastic modulus results differ according to the numerical analysis model and FEA. Generally, the longitudinal elastic moduli E2 and E3 are symmetrical under the assumption that the fiber arrangement is constant and are assumed to have the same value. Accordingly, the longitudinal elastic moduli used in this study were combined under E2. According to a comparison of the calculated longitudinal elastic moduli, the ROM model showed the greatest error with the FEA results, followed by modified ROM, Chamis, and Halpin–Tsai. These models can be analyzed using results that apply the correction value according to each model to reduce the longitudinal elastic modulus error.

In FEA, the elastic moduli in the E2 and E3 directions were symmetrical and did not have the same value because, as shown in Figure 1, the RVE system has a forward lattice whereas the fiber symmetric structure connecting the centers of the fibers is symmetrical in a rhombus shape; therefore, the longitudinal elastic moduli E2 and E3 are not symmetrical. If in Figure 2, ② were composed of a lattice in the same direction as RVE, then the longitudinal elastic modulus values would have been consistent.

Figure 3 shows the shear modulus calculated according to the volume fraction of UHMWPE. Similar to the elastic modulus results above, we conducted FEA up to a volume fraction of 78%. According to a comparative analysis of changes in shear modulus, calculated according to volume fraction through numerical analysis, the Chamis model showed the most similar results to FEA when the volume fraction of UHMWPE was less than 50%. The Halpin–Tsai model showed the most similar results to FEA at volume fractions of 60 to 70% but showed a rapid difference at volume fractions above 70%. As described above, this result can be attributed to the misalignment of symmetry between the RVE lattice and UHMWPE fibers.

Figure 4 presents a graph of changes in Poisson’s ratio according to the numerical analysis and FEA. According to the numerical analysis results, the changes in Poisson’s ratio according to the volume fraction of UHMWPE fibers were identical in the ROM, modified ROM, and Chamis models. Meanwhile, the Halpin–Tsai model exhibited the closest results to FEA at UHMWPE volume fractions below 40% but then converged with the existing micromechanics models as the volume fraction increased because, unlike other models, the Halpin–Tsai model calculates the values based on the shape and direction of the filler and the elastic properties of the filler and matrix. Furthermore, the FEA results indicate that the error of Poisson’s ratio increases as the volume fraction increases, unlike the existing numerical analysis models, suggesting that, just as differences occur among E1, E2, and E3 according to the fiber arrangement, the calculated values of Poisson’s ratio also vary with the symmetry of the fibers. Hence, as the volume fraction of UHMWPE fibers increases, Poisson’s ratio decreases, and the probability of the composite that exhibits a reduced strain to external forces increases.

We calculated the elastic moduli of the UHMWPE fiber-reinforced composite through numerical analysis and FEA. The transverse elastic modulus of the fiber linearly increased according to the volume fraction of UHMWPE whereas the longitudinal elastic modulus slightly differed according to the numerical analysis model. The error range of the Halpin–Tsai model and the longitudinal elastic modulus calculated through FEA and the homogenization method was the smallest. According to a comparison of the calculated shear modulus values, the FEA and numerical analysis results converged when the volume fraction of UHMWPE was below 70%. Therefore, by sufficiently mixing UHMWPE fibers and PP material with sufficient bonding force, it is possible to develop a composite of UHMWPE fibers with the reduced strain and reinforced elastic properties of PP.

## 3. Materials and Methods

To comparatively analyze the elastic properties according to the addition of UHMWPE fibers, we examined their changes according to volume fraction based on numerical analysis theory and FEA. The main advantage of micromechanics is that it enables virtual tests, which reduce the cost of experiments. In practice, experiments with heterogeneous materials are often costly and involve many permutations, such as constituent material combinations, fiber and particle volume fractions, fiber and particle arrangements, and treatment history. Once the component properties are known, these permutations can all be simulated in virtual tests using micromechanics. However, the micromechanics model must be verified by comparing experimental data other than those used in reverse engineering. In the numerical analysis, we mathematically calculated the values, considering the properties of UHMWPE and PP based on the conventional micromechanics model, whereas, in the FEA, we calculated the changes in elastic properties according to the volume fraction of fibers using the homogenization method. Table 1 shows the properties used in this study. In addition, the UHMWPE fiber used in this study is based on PM-200 of MIPELON™. We conducted finite element analysis and numerical analysis on PP by simulating the 90um powder of Goonvean-Fibers™ as a base material. The tensor notation used in this study is shown in Figure 5. In addition, the epithet “f” means fiber, and “m” means matrix.

### 3.1. Rule of Mixture (ROM)

A general ROM in materials science indicates the weighted average used to predict various properties of composites. ROM provides theoretical upper and lower limits for properties including elastic modulus, mass density, ultimate tensile strength, thermal conductivity, and electrical conductivity [19,20,21,22]. The simple ROM uses the Voigt and Reuss models with Equations (1)–(4) to predict the elastic properties of composites according to the volume fraction of the constituent materials. The Voigt and Reuss models are configured under the assumption of uniform strain and stress. Using E_11_ and ν_12_ (Voigt models) and E_22_ and G_12_ (Reuss models), the equations below can be configured. However, ROM is not an effective method for estimating tensile strength because increased stress and strain due to reinforcement, interfacial failure, statistical dispersion effects, voids, misalignment, and various other factors yield inaccurate tensile strength predictions. Though ROM may not be useful in the transverse direction of continuous fiber composites, it is frequently used because it is easy to derive linear results with the basic volume fraction. E_11_ is the longitudinal elastic modulus, E_22_ is the transverse elastic modulus, and G is the shear modulus.
(1)$$E_{11}=V^{f}E_{11}^{f}+V^{m}E^{m}$$
(2)$$E_{22}=\frac{E_{22}^{f}E^{m}}{E^{m}V^{f}+E_{22}^{f}V^{m}}$$
(3)$$\nu_{12}=V^{f}\nu_{11}^{f}+V^{m}\nu^{m}$$
(4)$$G_{12}=\frac{G_{12}^{f}G^{m}}{G^{m}V^{f}+G_{12}^{f}V^{m}}$$
$V^{f}$: Volume fraction of fiber$V^{m}$: Volume fraction of matrix$\nu_{11}^{f}$: Poisson’s Ratio of fiber$\nu^{m}$: Poisson’s Ratio of matrix

### 3.2. Modified ROM

Modified ROM is highly consistent with computational structural analysis data and longitudinal property experiments calculated with simple ROM but was improved by setting a correction factor for the transverse direction and shear properties as shown in Equations (5)–(8) [23,24,25,26]. The correction factor—the stress distribution coefficient—was empirically obtained through experimental data, through which improved properties can be obtained.
(5)$$\eta^{f}=\frac{E_{11}^{f}V^{f}+\left[\left(1-\nu_{12}^{f}\nu_{21}^{f}\right)E^{m}+V^{m}\nu_{21}^{f}E_{11}^{f}\right]V^{m}}{E_{11}^{f}V^{f}+E^{m}V^{m}}$$
(6)$$\eta^{m}=\frac{\left[\left(1-{\left(\nu^{m}\right)}^{2}\right)E_{11}^{f}-\left(1-\nu^{m}\nu_{21}^{f}\right)E^{m}\right]V^{f}+E^{m}V^{m}}{E_{11}^{f}V^{f}+E^{m}V^{m}}\sqrt{b^{2}-4ac}$$
(7)$$\frac{1}{E_{22}}=\frac{\eta^{f}V^{f}}{E_{22}^{f}}+\frac{\eta^{m}V^{m}}{E^{m}}$$
(8)$$\frac{1}{G_{12}}=\frac{\frac{V^{f}}{G^{f}}+\frac{\eta^{f}V^{m}}{G^{m}}}{V^{f}+\eta^{f}V^{m}}\;,\;(0<\eta^{f}<1)$$
$\eta^{f}$: Correction factor of fiber$\eta^{m}$: Correction factor of matrix

### 3.3. Chamis Model

The Chamis model is an analytical approach widely used by researchers to validate finite element result analysis to ensure that they can be productively employed for yielding the composites as shown in Equations (9)–(11). The Chamis model is a modified technique that substitutes the fiber volume fraction of the conventional simple ROM with the square root. Unlike the conventional ROM, it can measure the values of five effective properties, including G23, making this technique widely applicable [27,28,29,30].
(9)$$E_{22}=\frac{E^{m}}{1-\sqrt{V^{f}}(1-E^{m}/E_{22}^{f})}\;\;\;$$
(10)$$G_{12}=\frac{G^{m}}{1-\sqrt{V^{f}}(1-G^{m}/G_{12}^{f})}\;\;\;$$
(11)$$G_{23}=\frac{G^{m}}{1-\sqrt{V^{f}}(1-G^{m}/G_{23}^{f})}$$

### 3.4. Halpin–Tsai Model

Halpin and Tsai (1969) developed a semi-empirical method to predict composite properties. The Halpin–Tsai method sensibly interpolates between upper and lower bounds of composite properties. The Halpin–Tsai model is a mathematical model that predicts the elasticity of composites based on the shape and direction of the filler and the elastic properties of the filler and matrix [31,32,33,34,35]. In addition, as shown in Equations (12)–(15), this model uses the reinforcement factor (ξ) calculated through experiments to improve the existing ROM. The reinforcement factor depends on the shape of the fibers, their arrangement, and load conditions.
(12)$$\eta_{E}=\frac{(E^{f}/E^{m})-1}{(E^{f}/E^{m})+\xi }\;$$
(13)$$\eta_{G}=\frac{(G^{f}/G^{m})-1}{(G^{f}/G^{m})+\xi }\;$$
(14)$$E_{22}=E^{m}\left(\frac{1+\xi \eta V^{f}}{1-\eta V^{f}}\right)$$
(15)$$G_{12}=G^{m}\left(\frac{1+\xi \eta V^{f}}{1-\eta V^{f}}\right)$$
$\eta_{E}$: Elastic modulus correction factor$\eta_{G}$: Shear modulus correction factorη: Correction factorξ: Reinforcing factors

### 3.5. Finite Element Analysis Theory

It is difficult to numerically simulate composite structures due to the differences in the scale of the lengths, fibers, and matrix. FEA can be used to simulate the structural dynamics of a composite system but it is not practical. The homogenization technique is the standard approach to solve this size problem when performing FEA on composites [36,37,38,39]. ANSYS is used to define the micro-scale structure of the material in Material Designer and then FEA is performed based on this. In Material Designer, the homogenization process begins with modeling the RVEs, which requires the definition of the material properties and simplified shapes of the constituent materials before dividing the shapes into a finite number of elements for FEA. To conduct FEA in this study, we constructed models where UHMWPE fibers are uniformly arranged, as shown in Figure 6. An important variable in the numerical approach of composites is the volume fraction of the material. Accordingly, the RVE models were expressed such that the diameter of the UHMWPE fibers increases with the volume fraction of the increases. The most important variable in the micromechanics of composite materials is the volume fraction; while the change in composite properties according to differences in fiber diameter cannot be calculated, the changes in fiber diameter in the RVE can be ignored. In addition, fiber-containing composites are generally classified as anisotropic materials. The transverse elastic modulus of fiber-containing composites is large, but the longitudinal elastic modulus of the fibers is small; therefore, it is common to obtain anisotropic property values to perform the analysis when conducting FEA. However, the properties used in this study were the isotropic properties shown in Table 1. We modeled these isotropic properties such that anisotropic FEA could be conducted according to the shape of the fiber. FEA was performed while increasing the volume fraction of UHMWPE; according to the results, the maximum volume fraction of the UHMWPE fibers was 78%. The volume fraction limit in this numerical analysis was the same as the maximum theoretical limit of the volume fraction of a circle inscribed in a rectangle. Hence, the maximum volume fraction of UHMWPE fibers in the theoretical FEA was 78%, and thus, FEA was performed up to 78%.

## 4. Conclusions

In this study, we predicted the elastic properties of composites developed from mixing UHMWPE fibers and PP through numerical analysis and FEA. Using micromechanics models, we calculated the elastic properties according to the volume fraction of the composite depending on the amount of UHMWPE fibers added and comparatively analyzed the results via FEA. The results were as follows.

(1)The maximum volume fraction of the UHMWPE fibers that could be mixed was 78%, and it was possible to perform FEA using ANSYS Material Designer up to a volume fraction of 78%.(2)According to the micromechanics model and FEA results, owing to the high elastic properties of UHMWPE, the transverse elastic modulus of the composite increased linearly with the volume fraction of UHMWPE. In terms of longitudinal elastic modulus, the ROM model exhibited the largest error, while the Halpin–Tsai model showed the smallest.(3)According to a comparative analysis of shear modulus, the errors between the FEA and micromechanics models were small: up to 70% volume fraction of UHMWPE fiber.(4)The error that appeared as the volume fraction increased, and the difference in E2 and E3 longitudinal elastic modulus values from the FEA occurred because the fiber arrangement did not form a lattice in the forward direction with the RVE. If the fibers were not arranged in a rhombus shape but symmetrically in the forward direction, the difference in longitudinal elastic modulus and difference in shear modulus would not have been large.(5)According to calculations of Poisson’s ratio, the FEA results highly converged with the micromechanics models at UHMWPE volume fractions below 40%, whereas a large error occurred at volume fractions above 50%. Moreover, as the volume fraction of the UHMWPE fibers increased, Poisson’s ratio decreased, thus forming a composite with minimal deformation.

Based on these research results, additional studies are needed to manufacture composites that mix UHMWPE fiber and PP and conduct experiments and comparative analyses.

## Figures and Tables

**Figure 1 molecules-27-05752-f001:**
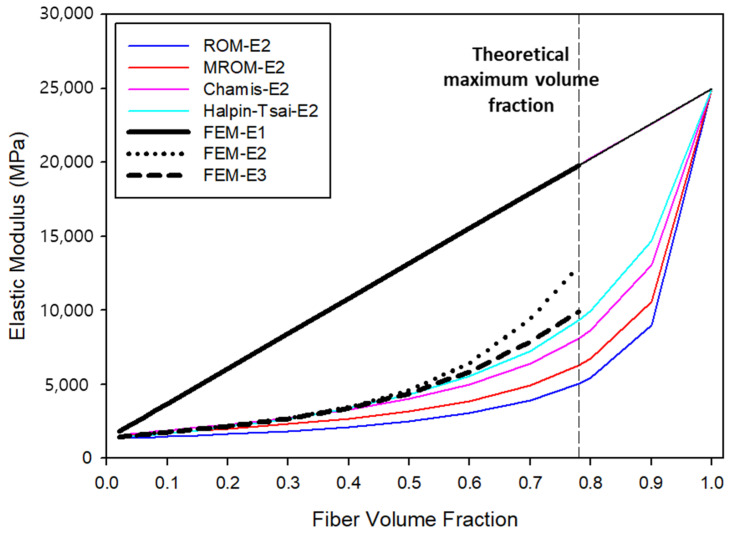
Changes in elastic modulus according to volume fraction of UHMWPE fibers.

**Figure 2 molecules-27-05752-f002:**
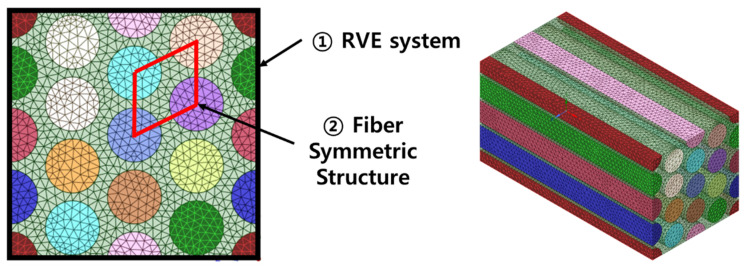
Arrangement of UHMWPE fibers.

**Figure 3 molecules-27-05752-f003:**
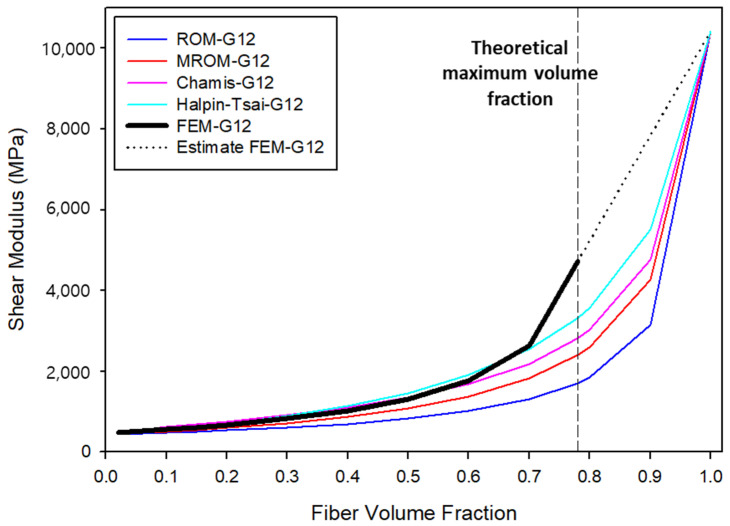
Changes in shear modulus according to volume fraction of UHMWPE fibers.

**Figure 4 molecules-27-05752-f004:**
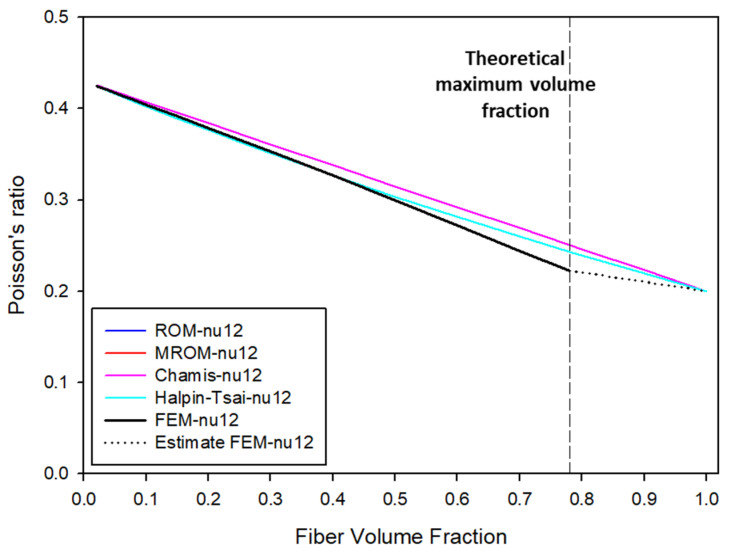
Changes in Poisson’s ratio according to volume fraction of UHMWPE fibers.

**Figure 5 molecules-27-05752-f005:**
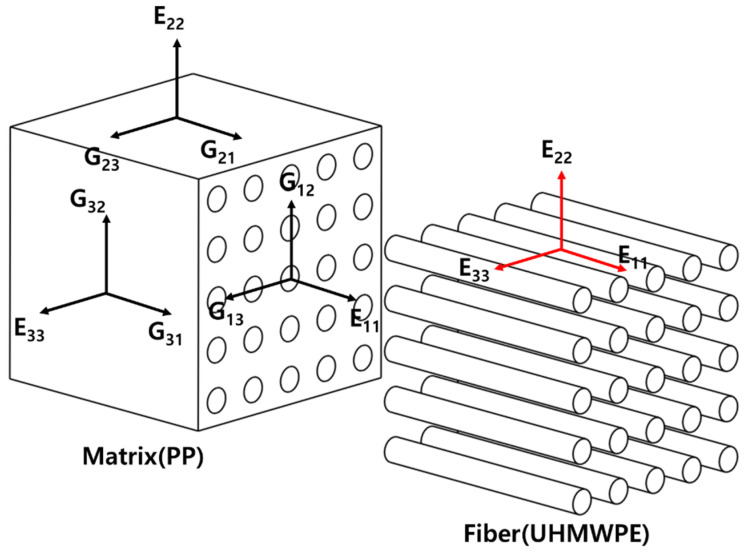
Tensor notation of composite.

**Figure 6 molecules-27-05752-f006:**
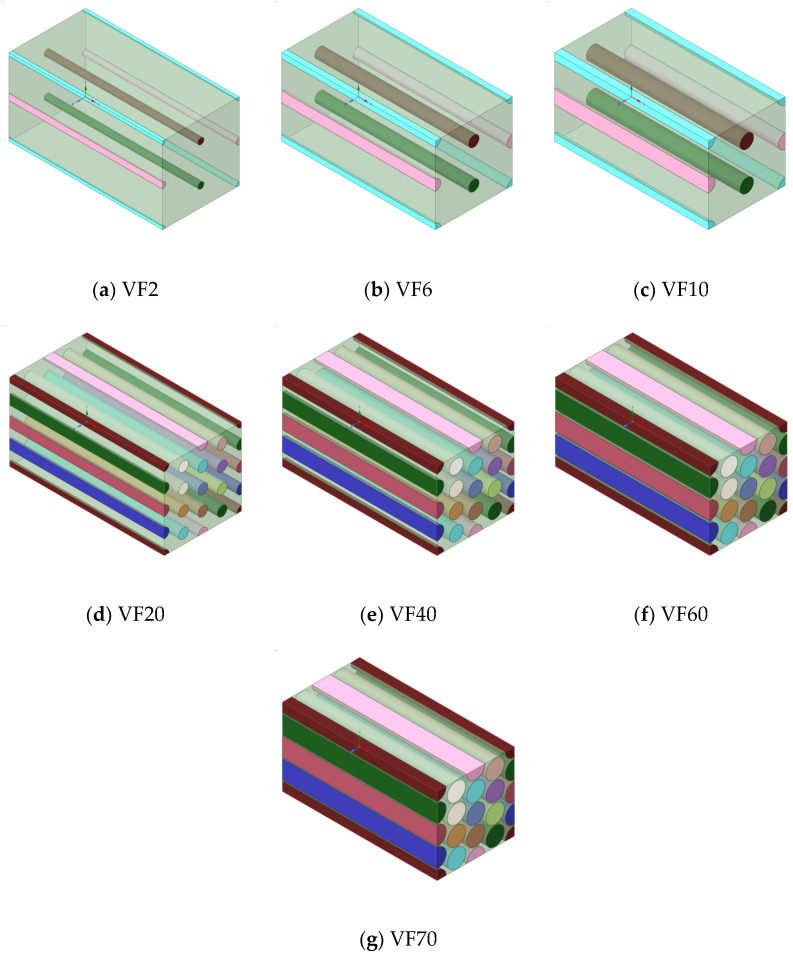
UHMWPE–PP composite FEA model by volume fraction (%). (**a**) 3D modeling at fiber VF 2%; (**b**) 3D modeling at fiber VF 6%; (**c**) 3D modeling at fiber VF 10%; (**d**) 3D modeling at fiber VF 20%; (**e**) 3D modeling at fiber VF 40%; (**f**) 3D modeling at fiber VF 60%; (**g**) 3D modeling at fiber VF 70%.

**Table 1 molecules-27-05752-t001:** Properties of PP and UHMWPE.

	Polypropylene (PP)	UHMWPE
**Elastic modulus (MPa)**	1325	25,000
**Shear modulus (MPa)**	432.29	10,417
**Poisson’s ratio**	0.43	0.20
**Bulk modulus (MPa)**	3154.8	13,889.0
**Density (kg/m^3^)**	904	950

## Data Availability

Not applicable.

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
