# Peer review of "Prediction of Elastic Properties Using Micromechanics of Polypropylene Composites Mixed with Ultrahigh-Molecular-Weight Polyethylene Fibers"

_molecules, 2022, doi:10.3390/molecules27185752_

Round 1

Reviewer 1 Report

The manuscript shows the calculus of material properties based on Ultrahigh-molecular weight Polyethylene Fibers. However, the manuscript lacks the mathematical formulation that allows support for the discussions of the predicted properties. I believe that the current manuscript is not suitable for the Molecules, despite the topic being aligned with the special issue. All comments are in the attached file.

Round 2

Reviewer 1 Report

The manuscript shows improvement contrasting the first version. However, some points need your attention. When carried out these changes, the manuscript can be accepted in Molecules. The comments are in the attached file.
